# Interfacial Charge Transfer Enhances Transient Surface Photovoltage in Hybrid Heterojunctions

**DOI:** 10.3390/nano15030154

**Published:** 2025-01-21

**Authors:** Cristian Soncini, Roberto Costantini, Martina Dell’Angela, Alberto Morgante, Maddalena Pedio

**Affiliations:** 1CNR—Istituto Officina dei Materiali (IOM), S.S. 14 km 163.5, 34149 Trieste, Italy; costantini@iom.cnr.it (R.C.); dellangela@iom.cnr.it (M.D.); morgante@iom.cnr.it (A.M.); pedio@iom.cnr.it (M.P.); 2Elettra Sincrotrone Trieste S.C.p.A., S.S. 14 km 163.5, 34149 Trieste, Italy; 3Dipartimento di Fisica, Università di Trieste, Via Valerio 2, 34127 Trieste, Italy; 4CNR—Istituto Officina dei Materiali (IOM), Via Pascoli, 06123 Perugia, Italy

**Keywords:** copper phthalocyanine heterojunctions, transient surface photovoltage, charge transfer, time-resolved photoemission spectroscopy

## Abstract

The interfacial energy level alignment in the copper phthalocyanine/SiO_2_/p-Si(100) heterojunction has been studied in dark conditions and under illumination. The element-sensitivity of the time-resolved X-ray photoemission provides a real-time picture of the photoexcited carrier dynamics at the interface and within the film, enabling one to distinguish between substrate and molecular contributions. We observe a molecule-to-substrate charge transfer under photoexcitation, which is directly related to the transient modification of the band bending in the substrate due to the surface photovoltage effect. Our results show that charge generation in the heterojunction is driven by the molecular layer in contact with the substrate. The different molecular orientation at the interface creates a new channel for charge injection in the substrate under photoexcitation.

## 1. Introduction

The quest for affordable and high-performance organic materials for photovoltaic applications relies on improving the ability to convert light into electrical current upon activation. However, increasing the efficiency of organic photovoltaic cells remains a significant challenge. When two semiconductors come into contact to form a heterojunction, the equilibration of the energy levels can introduce band bending and interfacial dipoles [1,2,3,4,5]. Molecule–substrate interactions can drastically impact the morphology and charge density distribution of the molecules at the interface [5,6,7]. Additionally, optical illumination may introduce transient modifications of the interfacial energy level alignment by photovoltaic effects and photoconductivity [4,8]. Therefore, a comprehensive understanding of photovoltaic devices must include a detailed description of the energy level alignment in both the ground state and transient conditions. To overcome the actual limits, recent research efforts have focused on two avenues: optimizing the efficiency of charge transfer (CT) between donor and acceptor materials and/or minimizing the loss mechanisms due to the recombination of the photo-induced carriers or excitonic states [9,10,11,12]. The former requires proper energy level alignment at the interface between materials, while the latter is influenced by the electronic properties and morphology of the interface.

Surface photovoltage (SPV) is a photo-induced effect that transiently modifies the energy level alignment at the interface in time scales ranging from sub-ps to s depending on the system [2,4,13,14,15]. SPV occurs whenever the system exhibits band bending, which is a common condition of semiconductor-based devices. In these systems, the excess of dopant charges is distributed over a screening length, generating an electrically non-neutral space charge region (surface electric field), and the bands in the region close to the interface are bent. Under illumination, the surface electric field splits the photo-excited species into free electrons and holes which are accelerated toward opposite directions, generating a voltage difference between the surface and bulk that lowers the band bending (SPV effect). In recent years, transient SPV spectroscopy [14,15,16] and time-resolved X-ray photoemission (TR-PES) studies have pointed out wide variability in SPV dynamics and recombination mechanisms due to a complex interplay between bulk and surface properties of the material [17,18,19,20].

The SPV effect is relaxed by electron–hole recombination at the surface after diffusion of one of the carriers from the bulk, following two mechanisms described in terms of the tunneling (fast process) and the thermionic relaxation (slow process) schemes. Initially, the depletion region is thin and bulk majority carriers can tunnel through the potential barrier to recombine with the excess surface charge. Since the potential barrier gradually restores (due to the electron–hole recombination), tunneling is no longer favorable and thermionic emission becomes the dominant mechanism. The hole diffusion to the surface slows down and the recombination efficiency decreases. From the existing literature, it is still not fully understood how the time scale of the SPV phenomenon is related to the different bulk (e.g., doping level) and surface parameters (e.g., surface termination). Surface defects (dangling bonds) are responsible for the presence of gap states strongly modulating the surface recombination velocity and the evolution in time of the SPV relaxation. Adsorption of atoms or molecules strongly tailors the band alignment, the carrier lifetime and charge transfer. The reduction in the number of dangling bonds by growing a layer on top of the semiconductor reduces the surface recombination, and CT between adsorbed species and the semiconductor has also been considered as a mechanism of SPV.

Here, we report a comprehensive study of molecule–substrate interaction effects on the interfacial energy level alignment and CT process in a CuPc/SiO_2_/p-Si(100) heterojunction by TR-PES. We focused on crystalline Si, which is the most employed photovoltaic material. We built p-n junctions by depositing copper phthalocyanine (CuPc) on silicon, a widely studied molecule for organic photovoltaic applications. Several studies, performed in dark conditions, pointed out a direct connection between CT efficiency and interface properties in CuPc-based heterojunctions [21,22,23,24]. Nevertheless, limited information is reported on the CT efficiency dependence on interfacial SPV-induced effects under illumination, which represents a further critical aspect that needs to be carefully evaluated. We focused on the transient change in the interfacial energy level alignment by the SPV. We observed CuPc-to-substrate CT under photo-excitation, which is well described by the measured transient modification of the band bending in the substrate. The CT is promoted only by photo-excited CuPc molecules of the first molecular layer. Molecule–substrate interaction modifies the CuPc electronic structure and film morphology at the interface, enabling the heterojunction’s charge separation and injection processes. Modeling of the TR-PES surface photovoltage effect data can be described in a quasi-equilibrium model based on charge carrier dynamics governed by thermionic emission (thermionic model). The model accurately reproduces the SPV dynamics before and after CuPc deposition, confirming that the observed modulation of the SPV signal is induced by modification of the transient charge density at the Si surface by CT from the molecular film.

## 2. Materials and Methods

All experiments were carried out at the ANCHOR-SUNDYN endstation [25] of the ALOISA beamline at the Elettra synchrotron in Trieste, Italy. Commercial CuPc powder (dye content > 99%) was purchased from Merck (Darmstadt, Germany). CuPc deposition was performed by thermal vapor deposition on a crystalline p-Si(100) wafer (B-doped 8 × 10^14^ cm^−3^) purchased from Siltronix (Munich, Germany). The substrate presented a thin oxide layer due to high purity dry oxidation (RCA treatment). RCA-treated Si showed a well-defined growth mode of the CuPc overlayers independently of the substrate conductivity [26]. The first layer preferentially adopts a flat-lying configuration and continuously rearranges to a standing geometry for increasing film thickness due to predominant molecule–molecule interaction. Despite the fact that an interfacial flat-lying geometry may favor hybridization between CuPc and substrate states (as in the case of metallic substrates), saturation of surface defects (Si dangling bonds) by CuPc adsorption over SiO_2_/Si has not been observed. Before CuPc deposition, the Si substrate was annealed at 450 °C in a base pressure in the range of 10^−9^ mbar to remove adsorbates on the surface due to air exposure. CuPc growth was monitored by in situ X-ray photoemission (PES) measurements. PES spectra were measured with a photon energy of 400 eV and an overall resolution of 0.15 eV. Energy calibration of the PES spectra was performed by acquiring the Si 2p core level with the first and second order of the synchrotron X-ray beam. In the TR-PES measurements, the optical laser and synchrotron beam were arranged in a quasi-collinear setup. The output of a Tangerine HP (Amplitude systems) operated at a 385 kHz repetition rate was frequency tripled at 343 nm. The fluence deposited on the sample was 0.5–15 µJ cm^−2^, focused to a diameter of 300 µm. The Elettra synchrotron was operated in hybrid mode, delivering a multi-bunch pulse pattern at 500 MHz with a separated X-ray pulse at a 1 MHz repetition rate. The acquisition was performed by separately measuring electrons emitted from both pumped and unpumped X-ray pulses, exploiting the different repetition rates between the pump and the probe as detailed in [27].

The SPV relaxation curves of the clean substrate and CuPc/SiO_2_/p-Si(100) heterojunction obtained by TR-PES were fitted following the thermionic model [13] (for more details refer to Appendix A):(1)$$SPV\left(t\right)=-\eta K_{B}T\ln\left(-\frac{1}{e^{\left[2V_{B}R_{bs}\left(\frac{t+\eta K_{B}T{SPV}_{0}}{N_{s}\eta K_{B}T}\right)-\ln\left(e^{\frac{{SPV}_{0}}{\eta K_{B}T}}-1\right)\right]}}+1\right)$$
where *N_s_* is the charge density at the surface, *V_B_* is the band bending, *R_bs_* is the bulk-to-surface flow of carriers under initial dark conditions, *SPV*(*t*) and *SPV*_0_ are, respectively, the SPV at a specific time delay and at saturation, *η* is the ideality factor, *K_B_* is the Boltzmann constant, and *T* is the temperature.

## 3. Results

### 3.1. PES

To understand the role of molecule–substrate interactions in the formation of the CuPc/SiO_2_/p-Si(100) interface, we first studied, using PES, the energy level alignment of the clean substrate and of the heterojunction at increasing CuPc coverages in dark conditions (without illumination). Figure 1a,b show Si 2p and C 1s core level lines as a function of the film thickness. The Si 2p core level has two main components, located at 99.2 eV and 103.5 eV of binding energy (BE), corresponding to the crystalline Si and SiO_2_ oxide layer, respectively. The double peak in the crystalline Si component is due to spin–orbit splitting (Si 2p_3/2_ and Si 2p_1/2_). The additional components in the fit correspond to the main SiO_2_ contribution and the weak sub-oxide species. The C 1s line shape can be accurately described using five Gaussian–Lorentzian components in the fit. The C_α_ and C_β_ components correspond to the inequivalent benzene and pyrrolic carbon rings, while S_α_ and S_β_ are the respective shake-up satellites [22,28,29]. The C_H_ component is assigned to the in-plane excitation of the C-H stretching mode [28,29]. Table 1 summarizes the relevant parameters of the C 1s fits for increasing CuPc coverage. The details of the fitting procedure are reported in the Appendix A.

From the Si 2p core level line of the clean substrate, we completely reconstructed the energy level alignment of the SiO_2_/p-Si(100) surface (Figure 1c). The crystalline Si shows a downward band bending of 170 meV. According to the charge neutrality condition, the excess of electrons in the Si bulk is counterbalanced by an equal amount of positive charges on the surface. However, the finite thickness of the oxide layer (1.2 nm) does not allow for efficient screening, and a net positive charging is induced at the SiO_2_/vacuum surface. The band bending and nominal thickness of the oxide layer were calculated using Appendix A.

The Si 2p and C 1s lines in Figure 1a,b show that upon increasing coverage up to 1.4 nm, the C_α_ and SiO_2_ peaks continuously shift towards higher BE, while the C_α_-C_β_ energy separation increases. Thicker films do not lead to any additional energy shift, and the C 1s line shape becomes comparable to the molecule in the bulk phase. Such behavior is in agreement with the growth on etched Si substrates found in ([26] and refs therein), suggesting the presence of molecule–substrate interaction effects during the formation of the interface (Appendix A). As the film thickness increases (bulk phase formation), molecules reorient toward the standing configuration, leading to increasing C_α_-C_β_ energy separation and decreased C 1s peak broadening (Table 1) [5,7,21,22,30], while the C_α_ and SiO_2_ peak shift towards the same direction points out the formation of an interface dipole [5,18]. We recall that the CuPc has a quadrupole moment, composed of a positive charge along the macrocycle plane and negative electron clouds below and above the ring, with a local electric field perpendicular to the molecular plane [31]. The positive charge on the SiO_2_/Si surface may induce rearrangement of the charge density on the macrocycle plane of the molecules directly in contact with the substrate, originating from the interface dipole^1^ and further directing the first CuPc layer in a flat-lying geometry [21,26]. In successive layers, molecules do not experience the interaction with the substrate, and the tilt angle continuously changes according to molecule–molecule interaction until stabilization in the standing mode. Therefore, the different polarizability of the environment for increasing coverage affects the C_α_-C_β_ energy separation.

### 3.2. TR-PES

We performed TR-PES measurements at various deposition stages, including on a clean substrate, in the low coverage regime, and after the first CuPc layer was completed (bulk-like film). We set the pump pulse energy to 3.61 eV (343 nm) to excite both materials simultaneously. We first evaluated the SPV on the clean substrate (Figure 2a). Upon excitation, the SPV causes a 95 meV shift in the Si 2p peak towards lower BE. The photo-saturation of the SPV is obtained at a pump fluence of 0.5 µJ cm^−2^, even if the theoretical value of the flat band condition is not reached (Appendix A). Since the SPV dynamics do not fully relax within the experimental time window (Figure 2c), we checked for pile-up effects, i.e., the observed SPV signal may be lower than the actual one. As discussed in the Appendix A, we observed a pile-up contribution to the experimental data of 20 meV (Appendix A). However, even considering such a decrease in the SPV signal, the saturation value does not coincide with the band bending. Several studies of Si systems report a similar behavior, but the origin of such discrepancy is not yet understood [17,18,19].

The crucial role of molecule–substrate interactions can be easily seen from the comparison between clean substrate and heterojunction SPV dynamics (Figure 2b,c). Upon CuPc adsorption, the Si 2p spectra show an enhancement of the SPV signal of 35 meV. After the completion of the first CuPc layer, we do not observe any additional change in the dynamics, suggesting that only molecules at the interface contribute to enhancing the SPV signal. Similarly to the clean substrate, the SPV saturation is achieved using a pump fluence of 0.5 µJ cm^−2^. Notably, the C 1s peak shows an identical energy shift. In the SPV process, any illumination-induced change in the energy levels in one material directly influences the energy levels of the material in contact [2]. Therefore, the C 1s peak directly reflects the SPV dynamics in the substrate, suggesting that the signal enhancement is thoroughly related to the SPV process. Such enhancement is compatible with the modification of the charge density at the Si surface via electron injection from the CuPc [32,33]. We verified the validity of a CT-induced enhancement of the SPV signal by fitting the dynamics response before and after CuPc deposition (Figure 2c) using the thermionic model (Equation (1)) [13]. Table 2 summarizes the best-fit parameters. The SPV dynamics after CuPc deposition are well reproduced only assuming a decreased hole charge density at the Si surface (*N_s_*) due to the screening of injected electrons from the CuPc. The difference between *N_s_* values before and after deposition represents the charge density transferred via CT. From the modeling, we obtained a CT density of 2 *×* 10^10^ cm^−2^, which is in good agreement with the calculated value of 1 *×* 10^10^ cm^−2^ (Appendix A) and compatible with the experimental photon fluence (2.3 *×* 10^12^ cm^−2^). The surface recombination velocity values obtained by fitting are comparable to those reported in the literature for oxidized Si surfaces [34]. As expected, upon CuPc deposition, the *S_v_* and *η* do not show significant changes. CuPc adsorption over the oxide layer does not affect the surface states at the crystalline Si side. We stress that the thermionic model does not account for organic interfaces and semiconductors. However, in our case, we can approximate the CuPc’s contribution to an additional source of charge density in the total balance of surface charges in the substrate. This approximation holds due to the inherent ultrafast nature of the singlet exciton CT in the CuPc [12,35] (<1 ps) as compared to the increase in the SPV (hundreds of ps), enabling us to introduce the CuPc film in the model simply by varying the initial *N_s_*.

## 4. Discussion

The molecule–substrate interaction induces a modification of the CuPc molecular orientation and of the electronic structure at the interface, creating a new channel for molecule-to-substrate CT. The thermionic model accurately reproduces the relaxation curves of the energy shifts before and after CuPc deposition, pointing out that the enhancement is fully related to SPV dynamics and induced by CT from the CuPc film. The identical C 1s and Si 2p TR-PES spectra after deposition, even in a CuPc bulk-like film of 5 nm (Figure 2c), confirm the absence of any additional contribution from CuPc intramolecular dynamics to the signal enhancement (if present, they are faster than the experimental time resolution). PES spectra rule out modification of the Si(100) energy levels upon CuPc deposition as a source of the increased SPV signal, i.e., the crystalline Si peak is unaffected for increasing coverages. After the completion of the first CuPc layer, thicker films do not lead to any further enhancement of the SPV signal, suggesting that only molecules directly in contact with the substrate contribute to the CT process. This fact agrees with the interface morphology (Appendix A) at the interface pointed out by PES data. A flat-lying configuration favors the coupling of CuPc orbitals at the interface with the substrate conduction band [21,23].

As shown in Figure 3, illumination generates photo-excited species in both materials. In the substrate, electrons and holes are separated and accelerated in opposite directions by the surface electric field (electrons migrate toward the surface). The potential difference between the surface and bulk induces the SPV signal of −95 meV. In the CuPc film, excitonic states relax into separated charges and electrons are injected into the Si, contributing to the surface–bulk potential difference in the substrate (SPV enhancement of 35 meV).

The first unoccupied states of the CuPc are perpendicularly oriented to the molecular plane [29]. We recall that in molecular crystals which self-assemble in long-range columnar structures (as in the CuPc case), charge transport and exciton diffusion are highly favored along the stacking axis [23,36,37,38,39]. Thus, the flat-lying geometry at the interface is highly favorable for charge delocalization toward the substrate, promoting the CT process, whereas the continuous change in the molecular stacking angle for increasing coverages results in a strong coupling of the CuPc unoccupied states between adjacent molecules (π-π interaction). As a consequence, intermolecular CT along the molecular stacking becomes the preferential path.

At the same time, the presence of an interface dipole may promote the dissociation of excitonic states in free charges, favoring the migration of electrons toward the oxide layer (positively charged)., This scenario is compatible with the observed saturation of the CT-induced enhancement in the low-coverage regime. Flat-lying geometry and interface dipole effects are limited to the interface. We stress that due to the modest entity of the interface dipole, we mainly address the driving force of the molecule-to-substrate CT to the interfacial flat-lying configuration which favors the coupling of CuPc and substrate empty states, enabling electron transfer via tunneling to the substrate.

## 5. Conclusions

The results presented here provide new insight into the photo-physics of hybrid CuPc/Si heterojunctions with a direct impact on the charge generation process by photoexcitation. The element sensitivity of the TR-PES technique provides a real-time picture of the photoexcited carrier dynamics at the interface and within the film, enabling us to distinguish substrate and molecular contributions. We observed an increase in the SPV signal upon photoexcitation due to CuPc-to-substrate CT. The CT originates only from photo-excited CuPc molecules of the first molecular layer thanks to a favorable interfacial morphology, while it is negligible in the upper layers due to the increasing tilt angle of the molecules, which hinders the mobility of the charges towards Si. Modeling of the TR-PES data by the thermionic model accurately reproduces the SPV dynamics before and after CuPc deposition, confirming that the observed modulation of the SPV signal is induced by modification of the transient charge density at the Si surface by CT from the molecular film.

## Figures and Tables

**Figure 1 nanomaterials-15-00154-f001:**
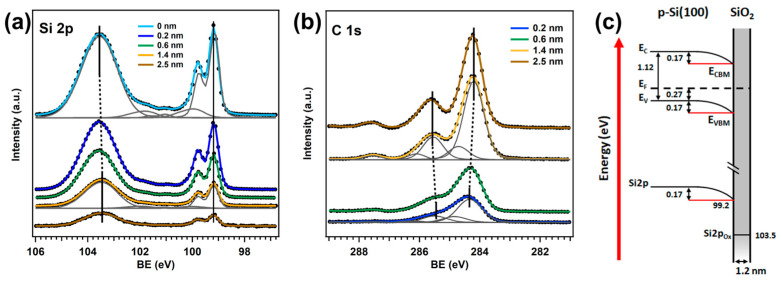
PES spectra (400 eV of photon energy) as a function of the film thickness of the (**a**) Si 2p and (**b**) C 1s core levels. Solid lines are the core level BE and dashed lines highlight the core level shifts as a function of the increasing film thickness; (**c**) the energy level model of the SiO_2_/p-Si(100) surface, reconstructed from the Si 2p core level of the clean substrate. The band bending and nominal thickness of the oxide layer were calculated using Appendix A.

**Figure 2 nanomaterials-15-00154-f002:**
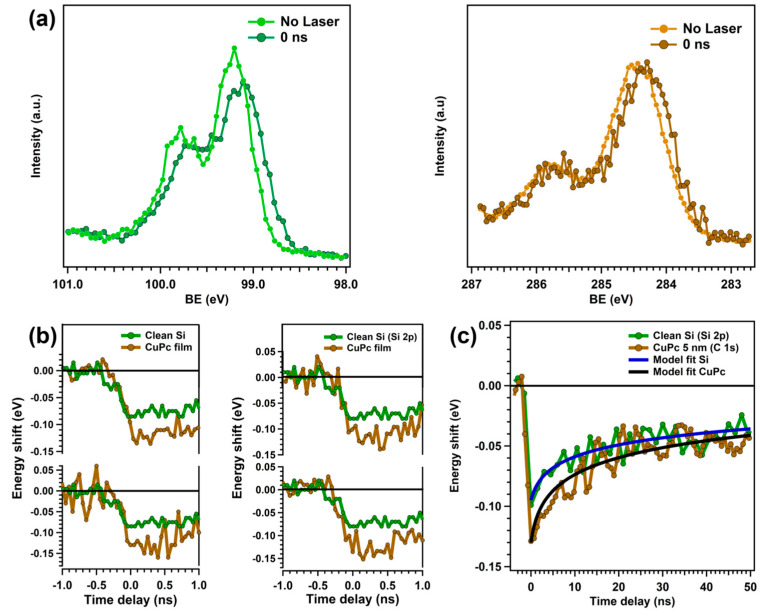
(**a**) Clean substrate (Si 2p core level) and heterojunction (C 1s core level) energy shifts upon laser excitation; (**b**) SPV dynamics after CuPc deposition at different film thicknesses (brown curves). The SPV relaxation curve of the clean substrate (green curves) is shown for comparison to highlight the enhancement of the SPV signal after CuPc deposition; (**c**) comparison between the clean substrate (Si 2p core level) and heterojunction (C 1s core level) SPV relaxation curves at longer time delays and the relative fits obtained by thermionic modeling.

**Figure 3 nanomaterials-15-00154-f003:**
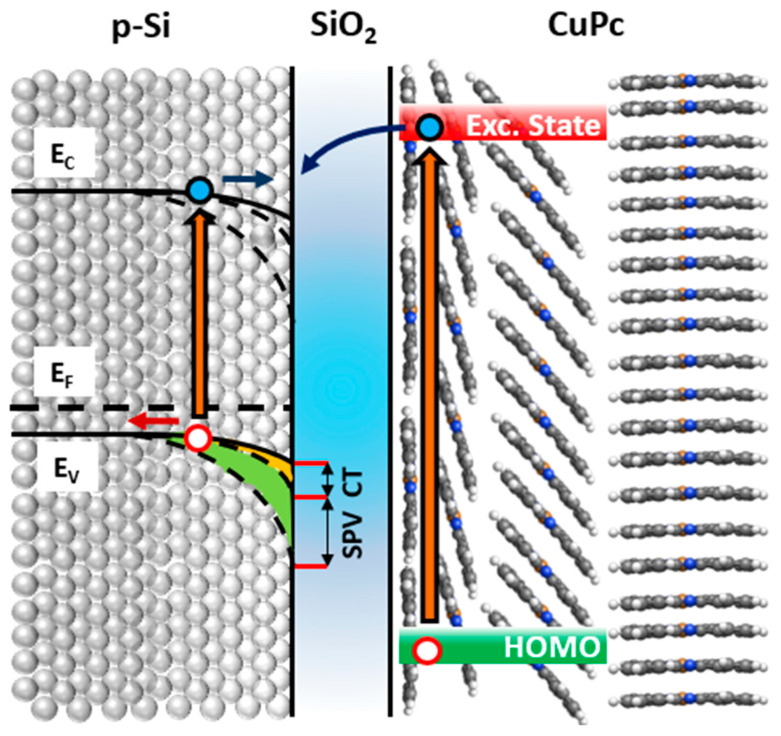
Proposed energy level model of the CuPc/SiO_2_/p-Si(100) heterojunction upon photoexcitation. The blue and red arrows show the direction of acceleration of photogenerated electrons and holes in the Si substrate, respectively.

**Table 1 nanomaterials-15-00154-t001:** Relevant parameters obtained by fitting the C 1s core level as a function of the film thickness. For more details on the fitting procedure, refer to the Appendix A.

Thickness (nm)	C_α_ BE (eV)	C_α_-C_β_ Energy Distance (eV)	FWHM (eV)
0.2	284.34	1.11	0.82
0.6	284.29	1.26	0.80
1.4	284.20	1.35	0.67
2.5	284.22	1.36	0.61

**Table 2 nanomaterials-15-00154-t002:** Best fit parameters obtained by thermionic modeling of the SPV relaxation curves before and after CuPc deposition (Figure 2c). *SPV*_0_ is the saturation value after laser excitation (t = 0), *N_s_* is the initial density of holes at the Si surface, *η* is the ideality factor and *S_v_* is the surface recombination velocity (for more details refer to Appendix A).

Sample	*SPV*_0_ (meV)	*N_s_* (cm^−2^)	*η*	*S_v_* (cm s^−1^)
SiO_2_/Si(100)	−95	4.23 × 10^10^	0.63	4870
CuPc/SiO_2_/Si(100)	−130	2.25 × 10^10^	0.8	4450

## Data Availability

Data are contained within the article and Appendix A.

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
