# Peer review of "Interfacial Charge Transfer Enhances Transient Surface Photovoltage in Hybrid Heterojunctions"

_nanomaterials, 2025, doi:10.3390/nano15030154_

Round 1
Reviewer 1 Report
Comments and Suggestions for Authors
In this paper, transfer enhances of carrier at the interface of heterojunction (copper phthalocyanine/SiO2/p-Si(100)) is reported. The results are interesting. The crystalline properties of each layer should be reported to verify the carrier transport process. It is suggested to use X-ray small Angle scattering spectroscopy to test the crystallization behavior of the film interface.
Author Response
Comment 1: In this paper, transfer enhances of carrier at the interface of heterojunction (copper phthalocyanine/SiO2/p-Si(100)) is reported. The results are interesting. The crystalline properties of each layer should be reported to verify the carrier transport process. It is suggested to use X-ray small Angle scattering spectroscopy to test the crystallization behavior of the film interface.
Response 1; We agree with the referee that SAXS would be a direct probe of the film's crystalline properties. However, the crystalline properties of CuPc thin films grown on several substrates (metals, crystalline Si, SiO2/Si, HOPG, etc..) have been extensively studied in the literature by structural (AFM, SEM, XRD – ref. 26 and refs therein of the main text) and spectroscopical methods (UPS, XPS, linearly polarized XAS – refs. 5-7,21,22,26 and refs therein of the main text), pointing out a direct relationship between molecular reorganization and modulation of the spectral features in XPS measurements at the interface. Furthermore, phthalocyanines are a well-known class of molecules which self-assemble in organized columnar stacking leading to highly ordered crystalline films. Since in our XPS spectra, we observe identical modulation of the core level peaks to the reported in the literature we are confident that the crystalline structure adopted by the molecule during the films’ growth is highly similar to previous studies. A detailed characterization of the layer-by-layer crystalline properties is interesting by itself, however, it is beyond the scope of this work. We better specified this point in the Section Materials and Methods.
Reviewer 2 Report
Comments and Suggestions for Authors
In this paper, the authors reported a comprehensive study of the molecule-substrate interaction effects on the interfacial energy level alignment and CT process in a CuPc/SiO2/p-Si(100) hetero-junction. And they confirm the charge generation in the heterojunction is driven by the molecular layer in contact with the substrate. This manuscript is well organized and presented clearly, therefore, I would like to recommend the manuscript published with the following minor revisions. Please check:
1.Besides XPS, other important characterizations such as SEM and XRD should be supplemented in the manuscript aiming at further confirming the successful preparation of the integrated system.
2.In the manuscript the authors claim “the interface dipole promotes the dissociation of excitonic states and the delocalization of electrons toward the oxide layer”, What are the additional ways to prove this?
3.In table 2, What does SPV0 stand for?
4. Some of the most recent references should be cited in the manuscript.
5. Reference format need to be corrected, for example Ref. 10, Ref. 28.
Author Response
Comment 1: Besides XPS, other important characterizations such as SEM and XRD should be supplemented in the manuscript aiming at further confirming the successful preparation of the integrated system.
Response 1: We agree with the referee that SEM and XRD would be direct probes of the crystalline properties of the film. However, the crystalline properties of CuPc thin films grown on several substrates (metals, crystalline Si, SiO2/Si, HOPG, etc..) have been extensively studied in the literature by structural (AFM, SEM, XRD – ref. 26 and refs therein of the main text) and spectroscopical methods (UPS, XPS, linearly polarized XAS – refs. 5-7,21,22,26 and refs therein of the main text), pointing out a direct relationship between molecular reorganization and modulation of the spectral features in XPS measurements at the interface. Furthermore, phthalocyanines are a well-known class of molecules which self-assemble in organized columnar stacking leading to highly ordered crystalline films. Since in our XPS spectra, we observe identical modulation of the core level peaks to the reported in the literature we are confident that the crystalline structure adopted by the molecule during the films’ growth is highly similar to previous studies. A detailed characterization of the layer-by-layer crystalline properties is interesting by itself, however, it is beyond the scope of this work. We better specified this point in the Section Materials and Methods.
Comment 2: In the manuscript the authors claim “the interface dipole promotes the dissociation of excitonic states and the delocalization of electrons toward the oxide layer”, What are the additional ways to prove this?
Response 2: We agree with the referee that our results don’t provide direct evidence that the interface dipole contributes to promoting the CT process. In particular, conductivity and electric field-induced second harmonic generation (EFISHG) measurements can directly clarify this point. Nevertheless, two samples with identical molecular geometry but different entities of the interface dipoles are required, which is not trivial.
In our case, XPS provides direct evidence of the presence of an interface dipole. Due to the modest dipole entity (0.1 V), we are not inferring that is the driving force of the observed CT process. In the main text, we suggest that the presence of an interface dipole is favourable to the excitonic dissociation generating free carriers. We mainly attribute the CT process to a more favourable molecular geometry at the interface (flat-lying geometry). We better clarify this point in the main text (Discussion section).
Comment 3: In table 2, What does SPV0 stand for?
Response 3: The typo has been amended
Comment 4: Some of the most recent references should be cited in the manuscript.
Response 4: We added additional references with more recent studies on the transient surface photovoltage effect (refs 14-16).
Comment 5: Reference format need to be corrected, for example Ref. 10, Ref. 28.
Response 5: The references have been corrected
Reviewer 3 Report
Comments and Suggestions for Authors
Hybrid heterojunctions of CuPc/SiO2/p-Si (100) were prepared and the study of the molecule-substrate interaction effects and the impact on the interfacial energy level alignment and charge transfer process were investigated using the technique of TR-PES (time-resolved photoemission spectroscopy).
I believe that the paper is in general ok, however, some things concerning the results and their interpretation are missing from this manuscript, in my opinion, as it follows:
1. In section regarding the preparation of the samples, there is nothing about the early stages of copper phthalocyanine adsorption which usually is characterized by the saturation of surface defects and by a flat lying disposition on the surface. Since these aspects are important in understanding the mechanisms of molecule-substrate interactions in the formation of CuPc/SiO2/p-Si interface, I think the authors should insert a comment regarding these things.
2. Since SPV signals depend not only on the charge transport efficiency but also on many other things, including light absorption, photo-generation, charge separation, recombination and trapping and emission of charge carriers, I think authors should insert some comments regarding these influences on SPV dynamics.
3. About the energy level model proposed in Figure 3, the “game” between the flat-lying angle geometry of molecules at the interface and the presence of interface dipoles is not sufficiently explained, since it has a crucial importance in improving the efficiency of CT and therefore the increase in SPV signals. I think the authors should insert more comments here.
Author Response
Comment 1: In section regarding the preparation of the samples, there is nothing about the early stages of copper phthalocyanine adsorption which usually is characterized by the saturation of surface defects and by a flat lying disposition on the surface. Since these aspects are important in understanding the mechanisms of molecule-substrate interactions in the formation of CuPc/SiO2/p-Si interface, I think the authors should insert a comment regarding these things.
Response 1: We inserted a comment regarding the early stages of CuPc deposition over SiO2/Si in the section Materials and Methods. (lines 96-103)
Comment 2: Since SPV signals depend not only on the charge transport efficiency but also on many other things, including light absorption, photo-generation, charge separation, recombination and trapping and emission of charge carriers, I think authors should insert some comments regarding these influences on SPV dynamics.
Response 2: We better commented on the influence of different parameters on the SPV dynamics (lines 53-68)
Comment 3: About the energy level model proposed in Figure 3, the “game” between the flat-lying angle geometry of molecules at the interface and the presence of interface dipoles is not sufficiently explained, since it has a crucial importance in improving the efficiency of CT and therefore the increase in SPV signals. I think the authors should insert more comments here.
Response 3: We better comment on this point (lines 273-292). In our case, XPS provides direct only direct evidence of the presence of an interface dipole. However, due to the modest dipole entity (0.1 V), we are not inferring that is the driving force of the observed CT process. In the main text, we suggest that the presence of an interface dipole is favourable to the excitonic dissociation generating free carriers. We mainly attribute the CT process to a more favourable molecular geometry at the interface (flat-lying geometry).